SciPost Physics

Submission

DESY-25-012

# How to Unfold Top Decays

Luigi Favaro[1,2], Roman Kogler[3], Alexander Paasch[4],
Sofia Palacios Schweitzer[1], Tilman Plehn[1,5] and Dennis Schwarz[6]

**1** Institut für Theoretische Physik, Universität Heidelberg, Germany
**2** CP3, Université catholique de Louvain, Louvain-la-Neuve, Belgium
**3** Deutsches Elektronen-Synchrotron DESY, Germany
**4** Institut für Experimentalphysik, Universität Hamburg, Germany
**5** Interdisciplinary Center for Scientific Computing (IWR), Universität Heidelberg, Germany
**6** Institute for High Energy Physics, Austrian Academy of Sciences, Austria

February 6, 2025

## Abstract

**Using unfolded top-quark decay data we can measure the top quark mass, as well as search for unexpected kinematic effects. We show how generative unfolding enables both tasks and how both benefit from unbinned, high-dimensional unfolding. Our method includes an unbiasing step with respect to the training data and promises significant advantages over standard methods, in terms of flexibility and precision.**

# 1 Introduction

Particle physics studies the fundamental properties of particles and their interactions, with the goal to discover physics beyond the Standard Model. The methodology is defined by the interplay between precision predictions and precision measurements. A key challenge is that perturbative quantum field theory makes predictions for partons, while experiments observe particles through their detector signatures. First-principle simulations link these two regimes [1]. They start with predictions for the hard process from a Lagrangian, and then add parton decays, QCD radiation, hadronization, and the detector response, to eventually compare with experimental data. This forward-simulation inference is the basis of, essentially, all LHC analyses.

The first problem with forward inference is that it requires access to the data and the entire simulation chain; neither of them are available outside the experimental collaborations. Second, it is not guaranteed that the best theory predictions are implemented in the forward simulation chain. Finally, in view of the high-luminosity LHC, hypothesis-driven forward analyses will overwhelm our computing resources for precision theory predictions and detector simulations. All three problems motivate alternative analysis techniques.

An exciting alternative analysis method is based on inverse simulations or unfolding. Instead of simulating detector effects for each predicted event, we can correct the observed events, for example, for detector effects. Then, we perform inference on particles before the detector or even partons and their hard scattering. Because the forward simulations are based on quantum physics and are stochastic, unfolding poses an incomplete inverse problem on a statistical basis. Still, in this way

1. analyses can be done outside the experimental collaborations;
2. theory predictions can be updated and improved easily;
3. and BSM hypotheses can be tested without full simulations.

Machine learning (ML) methods are revolutionizing not only our daily lives, but also LHC physics [2]. While classical unfolding methods are severely limited in many ways, ML-unfolding allows us to unfold unbinned events in many dimensions [3]. A reweighting-based ML-based unfolding method is MultiFold [4], applied to H1 [5–7], LHCb [8] and, recently, ATLAS [9] data. Generative ML-unfolding either maps distributions [10–14] or learns the underlying conditional probabilities [15–22]. Which of these complementary methods one would want to use depends on the specific task. Learning conditional probabilities to invert the forward simulation chain gives us access to per-event probabilities smoothly over phase space [23], guaranteeing the correct event migration. Its success rests on sufficiently precise generative networks [24–27], which are developed and benchmarked also for fast forward simulations [28–32].

In this paper we target an especially challenging unfolding task, mass measurements and the unfolding of strongly peaked kinematic distributions, applied to hadronic decays of boosted top quarks. It has been shown that it is possible to measure the top quark mass using matrix unfolding [33,34]. In Sec. 2 we describe the goal of the analysis, show the results from the classic CMS analysis, introduce the dataset, and sketch the basic features and the implementation of generative unfolding. In Sec. 3 we see how the top mass can be measured with unbinned unfolding. We find that a major problem is the bias induced by the training data. It can be ameliorated as described in Sec. 3.2. Next, we show in Sec. 3.3 how the top mass can be measured from the unfolded distributions, and in Sec. 3.4 we show how to then unfold the entire top decay phase space for re-analysis. The goal of this paper is to show that decay kinematics can be unfolded and to provide a blueprint for an LHC analysis using generative unfolding.

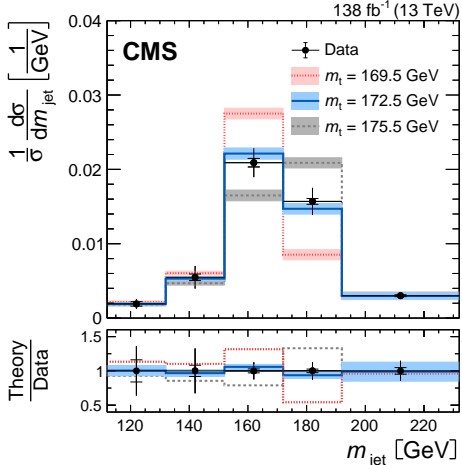

Figure 1: CMS benchmark result from Ref. [34]. It shows the differential top pair cross section as a function of the top-jet invariant mass, compared to theory predictions for different top masses. The vertical bars represent the total uncertainties, statistical uncertainties are shown as short horizontal bars, and theoretical uncertainties as shaded bands.

## 2 Goal and method

If we want to unfold top-quark decay events, the main challenge is the model dependence and resulting bias when the top masses assumed for the simulated training data and the actual top mass differ. This could be taken care of with iterative improvements of the unfolding network, but it will turn out that this approach is extremely challenging numerically. Instead, we follow a slightly different strategy:

1. we ensure that the bias from the top mass assumed in the simulated training data is small;
2. we infer the correct top mass from the data, using a reduced unfolded phase space;
3. we produce training data with the inferred top mass and unfold the full phase space.

### 2.1 Top mass measurement

The extraction of the top mass from the invariant jet mass of highly boosted hadronic top quark decays can shed light on possible ambiguities in top mass measurements using simulated parton showers. The ultimate goal is to compare the measured jet mass distribution to predictions from analytic calculations. For that, it is convenient to unfold detector effects.

Unfolding uses simulated data, biasing the unfolded data towards the model used in the simulation. In particular, the choice of the top mass in the simulation leads to a significant uncertainty [34]. These modelling biases can be reduced by including more information and granularity into the unfolding process, motivating the use of ML-unfolding methods.

In the existing CMS measurement this is done by also unfolding differentially in the top-jet transverse momentum and by including various sideband regions close to the measurement phase space. Using ML-unfolding, the data can be unfolded in a larger number of phase space dimensions, providing ways to reduce the model bias.

The result from our CMS benchmark analysis [34] is shown in Fig. 1. This analysis unfolds the reconstructed 3-subjet mass $M_{jjj}$ and the corresponding reconstructed transverse momentum, $p_{T,jjj}$ to measure the top mass. The three subjets are obtained using a two-step clustering with the eXclusive Cone (XCone) algorithm [35]. In the first step, the event is clustered into

two large-radius jets with a distance parameter $R = 1.2$ to capture the decay products of the top quark and antiquark. In a second step, the two large-radius jets from the first step are each reclustered into three XCone subjets with $R = 0.4$, where the subjets capture the dynamics of the hadronic top quark decay. Before the unfolding, the jet mass scale is calibrated by reconstructing the $W$-boson from the two light-quark subjets and fitting the subjet energy scales to the resulting $W$ peak. The $W$ boson decay is identified with the help of $b$-tagging information, which is obtained for the XCone subjets by matching these in angular distance to small-$R$ anti-$k_T$ jets. This matching is needed because the $b$-tagging information is not calculated for XCone subjets in CMS. The uncertainty in the unfolding from the modeling of final state radiation is reduced with the help of another auxiliary measurement of $N$-subjettiness ratios [36] on large-$R$ anti-$k_T$ jets, matched by angular closeness to the large-$R$ XCone jets. The matching procedures and auxiliary measurements add considerable complications to the measurement and come with non-negligible uncertainties. Because of the finite efficiency of the $b$ tagging and the associated mis-identification rate, the information from the $W$ reconstruction cannot be used in the unfolding because it breaks the permutation invariance among the jets. The leading systematic uncertainties in this measurement originate from the jet energy scale, jet mass scale, jet mass resolution, the $b$-jet response and the unfolding bias from the choice of the top mass in the simulation. Non-negligible uncertainties also arise from the modeling of non-perturbative effects. Ideally, unfolding enough phase space dimensions to capture the $W$ decay and the salient features of the jet substructure should allow us to constrain the dominating uncertainties in-situ and remove the top-mass bias in the unfolding.

Once we have measured the jet mass in an event sample and consequently the top mass, we can further analyze the unfolded dataset. For instance, we can look for effects from higher-dimensional SMEFT operators on the decay of boosted tops, or we can search for anomalous kinematic distributions from new particles, modified interactions, or enhanced QCD effects at the subjet level. While the unfolding for the top mass measurement has to include a sufficiently large number of dimensions, as discussed above, we now need to unfold the full, 12-dimensional phase space. Three of these dimensions are finite jet masses, generated by QCD effects.

## 2.2 Dataset

We use simulated events for top pair production, similar to the one used for the CMS jet mass measurement [34]. We generate the events with Madgraph 5 [37]. Hadronization, parton showers, and multiple parton interactions are simulated with Pythia 8.230 [38] with the underlying event tune CP5 [39]. The samples include a simulation of the detector response implemented in Delphes 3.5.0 [40] using the default CMS card with pile-up, and the e-flow algorithm. The pile-up subtraction only removes charged tracks associated to pile-up vertices.

In the simulated data, we have access to three stages of the simulation chain. The parton level includes the hard interactions of the top quarks, that decay into a $b$-quark and a $W$-boson, that subsequently decays into two quarks or lepton and neutrino. The particle level refers to all stable particles with lifetimes longer than $10^{-8}$ s after parton shower and hadronization. Finally, the detector level describes particle candidates after the detector simulation.

Event selections are applied at the particle and detector level. All events that do not pass either of the selections are rejected from further analysis. For the signal or measurement region, we only consider $t\bar{t}$ pairs in the lepton+jets decay at the parton level,

$$pp \to t\bar{t} \to (bq\bar{q}')\,(\bar{b}\ell^-\bar{\nu}) + \text{c.c.} \quad \text{with} \quad \ell = e, \mu\,, \tag{1}$$

with the lepton acceptance

$$p_{T,\ell} > 60 \,\text{GeV} \quad \text{and} \quad |\eta_\ell| < 2.4 \,. \tag{2}$$

The top jet is constructed using XCone clustering and identified by the larger angular distance to the lepton. It must fulfill

$$p_{T,J} > 400 \,\text{GeV} \qquad \text{and} \qquad p_{T,j_{1,2,3}} > 30 \,\text{GeV} \qquad |\eta_{j_{1,2,3}}| < 2.5 \,, \tag{3}$$

for the large-$R$ jet $J$ and three subjets $j_i$. In the following, we will refer to these subjets as jets. The second large-$R$ jet has to have $p_{T,J} > 10 \,\text{GeV}$ to reject poorly reconstructed events where only the lepton and not the $b$ quark is reconstructed in the second large-$R$ jet. To reduce the contribution from events where the full top quark decay is not reconstructed within the top jet, we require the invariant mass of the three jets, $M_{jjj}$, to exceed the invariant mass of the lepton and the large-$R$ jet close to it.

At the detector level, in addition to the above requirements, the missing transverse momentum has to be larger than $50 \,\text{GeV}$ and at least one $b$-tagged jet must be present.

The measurement-region selection criteria leave us with approximately 800,000 events simulated with a top mass of $m_t = 172.5 \,\text{GeV}$, of which we use 75% for the training. To be consistent with the amount of events available in CMS with the full detector simulations for the reference analysis, we choose samples with different top masses to have less events. All events contain the full generator (gen) and recoconstruction (reco) level information. The XCone algorithm clusters the jets separately for reco-level jets and gen-level jets. The clustered jets are sorted according to $p_T$.

## 2.3 Jet-mass features

For the generative unfolding algorithm a perfect matching between reco-level and gen-level jets is not critical, as the reco-level is used only as a condition. We have checked that when permuting the ordering of the reco-level jets randomly, we observe no difference in performance. Once we switch to the 4-momentum representation $(m, p_T, \phi, \eta)$, we see small differences between reco-level and gen-level, for instance in the $p_T$ and individual jet masses shown in Fig. 2 (top row).

Differences in the jet masses are mostly due to pile-up in our simulation, which is added at the reco-level, and to a lesser degree from inefficiencies and mis-reconstructions in the reconstruction of photons, charged and neutral hadrons. Pile-up contributions are reduced by removing tracks originating from pile-up vertices. The remaining difference in the jet mass mostly comes from photons and neutral hadrons in the pile-up. This positive contribution to the jet masses is largest for the leading jet because of its larger $p_T$ compared to the other jets. Figure 2 implies that unfolding detector effects includes correcting for these pile-up effects. As Delphes assumes an idealized vertex reconstruction, we expect those differences to be larger when including full detector effects with GEANT4 [41].

Going beyond single-jet observables, we need to understand and eventually unfold detector effects on jet-jet correlations. In Fig. 2 (middle row) we show two examples. The distribution in the angular separation between the two leading jets shows a characteristic peak, originating from the boosted decay kinematics combined with mass effects and the detector acceptance. The 2-jet masses have a peculiar distribution, owed to the fact that out of the three jets two come from the $W$ decay. Because of the $p_T$-ordering, any of the three combinations

$$M_{ik}^2 = m_i^2 + m_k^2 + 2 \left( m_{T,i} m_{T,k} \cosh \Delta y_{ik} - p_{T,i} p_{T,k} \cos \Delta \phi_{ik} \right) \tag{4}$$

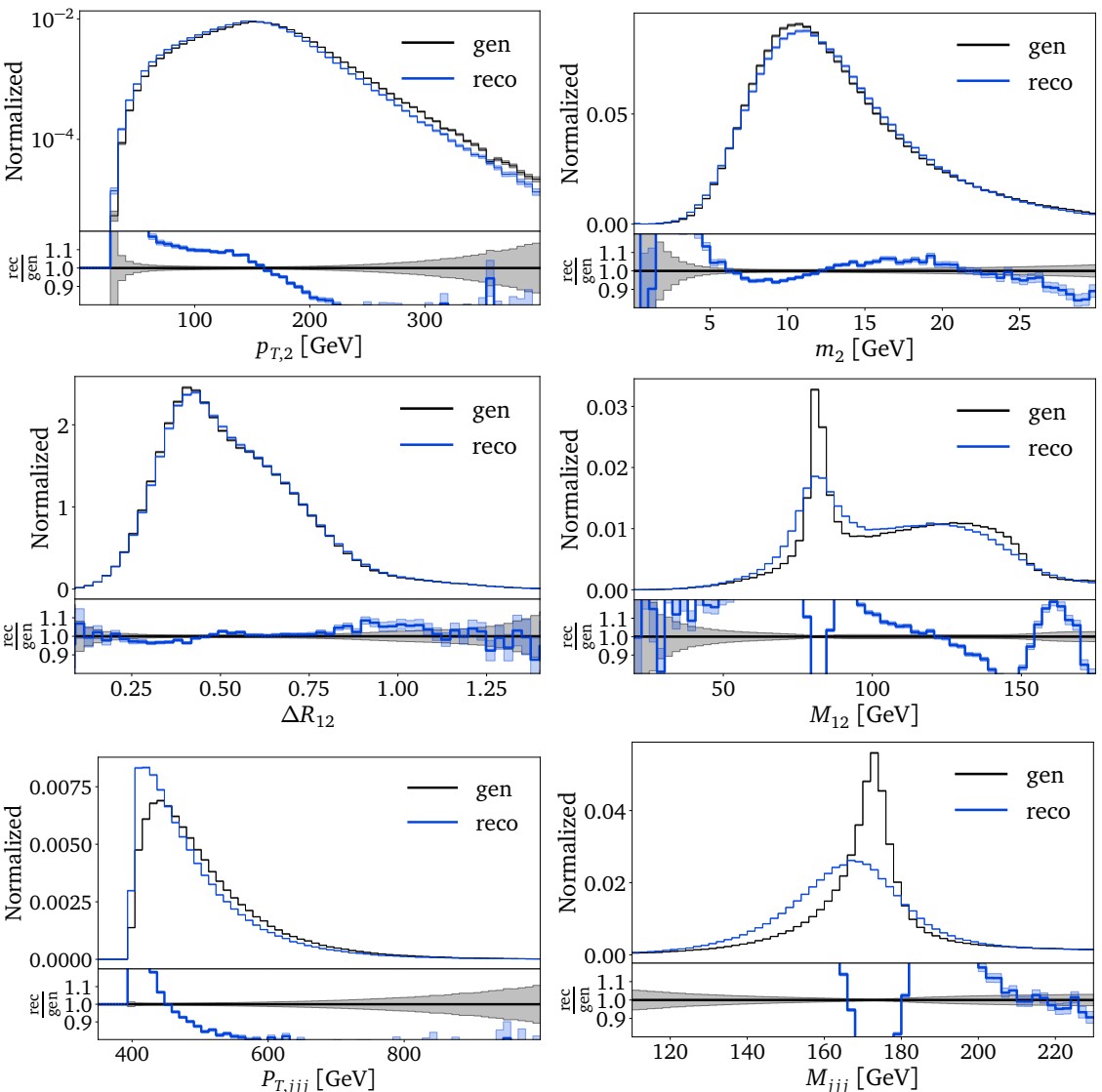

Figure 2: Kinematic distributions at reco-level and gen-level for the second jet (top), combining two jets (center), and combining three jets (bottom).

can reconstruct $m_W$. This is an exact equation for the three 2-jet masses, where $\Delta y_{ik}$ represents the difference in jet rapidities. Of the three 2-jet masses in a top decay, two tend to be similarly close to $M_{ik} \sim m_W$ [42]. In Fig. 2 (middle right), we also observe the upper endpoint in the top decay kinematics at gen-level [43]

$$m_{bj}^{\max} < \sqrt{m_t^2 - m_W^2} \approx 155\,\text{GeV}\,. \tag{5}$$

Following Eq.(4), we can improve the training of the unfolding network by including the 2-jet masses as explicit features. Each of the 2-jet masses then substitutes an angular variable. With this basis transformation we sacrifice access to the individual azimuthal angles and are left with their absolute differences.

Next, we see in Fig. 2 (bottom row) that the transverse top quark momentum is not affected significantly by detector effects, and the 3-jet mass peaks around the top mass value. In our phase space parametrization we can calculate the 3-jet mass as

$$M_{jjj}^2 = M_{12}^2 + M_{23}^2 + M_{13}^2 - m_1^2 - m_2^2 - m_3^2. \tag{6}$$

By using all these jet masses as training features, we can greatly improve the learning and unfolding of the 3-jet mass. The no-free-lunch theorem, however, tells us that this gain will lead to a mismodelling of other correlations. In particular, we will see that there is no guarantee that $\cos \Delta \phi \in [0, 1]$ anymore, leading to the generation of unphysical event kinematics in some cases.

## 2.4  Generative unfolding

Traditional unfolding algorithms [44–46] have been used to unfold simple differential cross section measurements. Widely used methods include Iterative Bayesian Unfolding [47–50], Singular Value Decomposition [51], and TUnfold [52]. Their limitation is the need for binned data in a low-dimensional phase space. This also means that we have to preselect the observables we want to unfold and decide on their binning before the unfolding.

To use ML-methods for high-dimensional and unbinned unfolding, we invert the forward simulation using Bayes' theorem

$$p(x_{\text{gen}}|x_{\text{reco}}) = p(x_{\text{reco}}|x_{\text{gen}}) \, \frac{w(x_{\text{gen}})p(x_{\text{gen}})}{w(x_{\text{reco}})p(x_{\text{reco}})} \, , \tag{7}$$

where $x_{\text{gen}}$ is a point in the weighted gen-level phase space and $x_{\text{reco}}$ a point in the weighted reco-level phase space. To unfold reco-level data, we need to learn

$$p_{\text{model}}(x_{\text{gen}}|x_{\text{reco}}) \approx p(x_{\text{gen}}|x_{\text{reco}}) \tag{8}$$

as the statistical basis of an inverse simulation. Once a generative neural network encodes $p_{\text{model}}(x_{\text{gen}}|x_{\text{reco}})$, we calculate

$$p_{\text{unfold}}(x_{\text{gen}}) = \int dx_{\text{reco}} \, p_{\text{model}}(x_{\text{gen}}|x_{\text{reco}})w(x_{\text{reco}})p(x_{\text{reco}}) \, . \tag{9}$$

At the event level, this integral can easily be evaluated by marginalizing the corresponding joint probability. Our method can be summarized as

$$
\begin{array}{ccc}
p_{\text{sim}}(x_{\text{gen}}) & & p_{\text{unfold}}(x_{\text{gen}}) \\
\text{paired data} \Big\updownarrow & & \Big\uparrow {\scriptstyle p_{\text{model}}(x_{\text{gen}}|x_{\text{reco}})} \\
p_{\text{sim}}(x_{\text{reco}}) & \xleftrightarrow{\text{correspondence}} & p_{\text{data}}(x_{\text{reco}}) \, .
\end{array}
\tag{10}
$$

The two distributions $p_{\text{sim}}(x_{\text{reco}})$ and $p_{\text{sim}}(x_{\text{gen}})$ are encoded in one set of simulated events, before and after detector effects, or at the parton- and the reco-level.

The generative network we employ to learn $p_{\text{model}}(x_{\text{gen}}|x_{\text{reco}})$ is Conditional Flow Matching (CFM). The generative CFM network is the leading architecture for precision-LHC simulations [26]. Mathematically, CFM is based on two equivalent ways of describing a diffusion process using an ordinary differential equation (ODE) or a continuity equation [53]

$$\frac{dx(t)}{dt} = v(x(t), t) \qquad \text{or} \qquad \frac{\partial p(x, t)}{\partial t} = -\nabla_\theta \left[ v(x(t), t)p(x(t), t) \right] \, , \tag{11}$$

both with the same velocity field $v(x(t), t)$. The diffusion process described by $t \in [0, 1]$ relates a Gaussian $r$-distribution to the physical phase space $x$,

$$p(x, t) \rightarrow \begin{cases} p_{\text{data}}(x) & t \rightarrow 0 \\ p_{\text{latent}}(r) = \mathcal{N}(r; 0, 1) & t \rightarrow 1 \, . \end{cases} \tag{12}$$

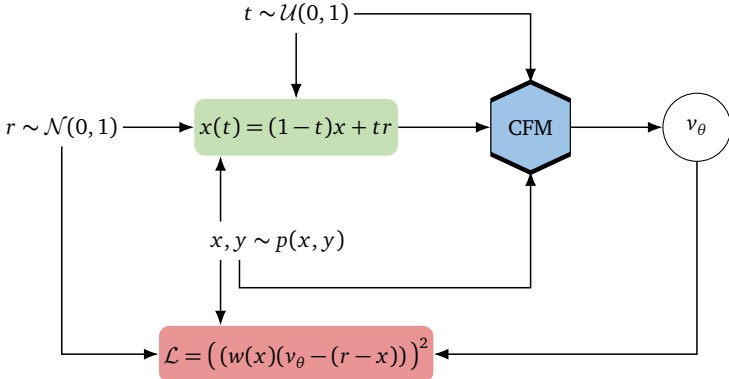

Figure 3: Schematic representation of generative unfolding with a CFM network.

We employ a simple linear interpolation

$$x(t) = (1-t)x + tr \rightarrow \begin{cases} x & t \rightarrow 0 \\ r \sim \mathcal{N}(0,1) & t \rightarrow 1 \,. \end{cases} \tag{13}$$

Using this approximation, we train the network to learn

$$v_\theta(x(t), t) \approx v(x(t), t) \tag{14}$$

using the continuity equation and then generate phase space configurations using a fast ODE solver. Even though the corresponding MSE loss function

$$\mathcal{L}_{\text{CFM}} = [w(x)(v_\theta - (r - x))]^2 \tag{15}$$

is not a likelihood loss, a Bayesian version of the CFM generative network can learn uncertainties on the underlying phase space density together with the central values underlying its sampling [26].

The CFM setup is illustrated in Fig. 3. Its conditional extension is straightforward, in complete analogy to the conditional GANs [15] and conditional INNs [16] developed for unfolding. While the naive GAN setup does not learn the event-wise (inverse) migration correctly and therefore does not encode physical, calibrated conditional probabilities, the cINN with its likelihood loss does exactly that. The CFM succeeds because of its mathematical foundation, Eq.(11) [3].

**Training bias**

In Eq.(10) we describe the structure of generative unfolding, but we are missing a critical complication — the simulated reco-level data $p_{\text{sim}}(x_{\text{reco}})$ might not agree with the actual reco-level data $p_{\text{data}}(x_{\text{reco}})$. Let us assume a simple case where the simulation depends on a simulation parameter $m_s$ which we can tune to describe the actual data. This can be a physics parameter we eventually infer, or a nuisance parameter which we profile over. The dependencies of the four datasets on $m_s$ and its 'correct' value in the data, $m_d$, turn Eq.(10) into

$$\begin{array}{ccc} p_{\text{sim}}(x_{\text{gen}}|m_s) & & p_{\text{unfold}}(x_{\text{gen}}|m_s, m_d) \\ {\scriptstyle p(x_{\text{reco}}|x_{\text{gen}})}\Big\downarrow & & \Big\uparrow{\scriptstyle p_{\text{model}}(x_{\text{gen}}|x_{\text{reco}}, m_s)} \\ p_{\text{sim}}(x_{\text{reco}}|m_s) & \xleftrightarrow{\text{correspondence}} & p_{\text{data}}(x_{\text{reco}}|m_d) \,. \end{array} \tag{16}$$

In the forward direction, $p(x_{\text{reco}}|x_{\text{gen}})$ does not have an explicit $m_s$-dependence, but both simulated datasets follow $p_{\text{sim}}(x_{\text{gen}}|m_s)$ and $p_{\text{sim}}(x_{\text{reco}}|m_s)$ induced by the generator settings. By assumption, $m_s = m_d$ ensures that the simulated and actual data agree at the reco-level,

$$p_{\text{sim}}(x_{\text{reco}}|m_s = m_d) \overset{!}{=} p_{\text{data}}(x_{\text{reco}}|m_d) \,. \tag{17}$$

We then use this relation to infer $m_d$ at the reco-level.

Alternatively, we can do the same inference at the gen-level, requiring

$$p_{\text{sim}}(x_{\text{gen}}|m_s = m_d) \overset{!}{=} p_{\text{unfold}}(x_{\text{gen}}|m_s = m_d, m_d) \,. \tag{18}$$

The problem with this unfolded inference is the dual dependence of $p_{\text{unfold}}(x_{\text{gen}}|m_s, m_d)$ through the reco-level data and the learned conditional probability. This dual dependence is automatically resolved if $p_{\text{unfold}}(x_{\text{gen}})$ only depends on $m_d$ through the reco-level data, so the bias from $p_{\text{model}}(x_{\text{gen}}|x_{\text{reco}}, m_s)$ can be neglected. If not, we can use iterative methods [18] to remove the bias. The iterative improvement relies on a learned classifier over $x_{\text{gen}}$ which reweights $p_{\text{sim}}$ to $p_{\text{unfold}}$ including the $m_s$-dependencies and serves as a basis for re-training the unfolding network. It implicitly assumes that $p_{\text{unfold}}(x_{\text{gen}}|m_s, m_d)$ depends mostly on $m_d$ and at a reduced level on $m_s$. In that case the endpoint of the Bayesian iteration is reached when the two dependencies coincide at the level of the remaining statistical uncertainty.

## 3 Unbinned top quark decay unfolding

Unfolding top decays is technically challenging, because the top mass and the $W$ mass are dominant features of an altogether 12-dimensional phase space. We start with a naive unfolding in Sec. 3.1, using our appropriate phase space parametrization with reduced dimensionality [20]. In Sec. 3.2, we show how the model dependence from the top mass in the training data can be controlled. With this enhancement, we show in Sec. 3.3 how the high-dimensional unfolding improves the existing top mass measurement based on classic unfolding. Finally, we show how to unfold the entire 12-dimensional phase space using the measured top mass in Sec. 3.4.

### 3.1 Lower-dimensional unfolding

We know that the precision of learned phase space distribution using neural networks scales unfavorably with the phase space dimension [54, 55].* The full 12-dimensional phase space will not be the optimal representation to measure the top mass. Instead, we only use a lower-dimensional phase space representation for the top mass measurement, finding a balance between relevant kinematic information and dimensionality. We postpone the full kinematic unfolding to the point where we need to access the full kinematics and benefit from the measured top mass.

For the traditional CMS analysis [34], two phase space dimensions were unfolded, $M_{jjj}$ and $p_{T,jjj}$, where the $p_{T,jjj}$ was integrated over in the final measurement. The jet mass calibration relies on the reconstructed $W$ boson. Identifying the $W$-decay jets in the top jet ideally requires $b$-tagging information, but because of the finite efficiency not all jets from the $W$ decay can be identified. Instead, the jet mass can be calibrated by using all possible 2-jet combinations, where each of the three resulting distributions feature a sharp $W$-mass peak (see Fig. 2). Therefore, we unfold those for the top mass measurement such that a reliable calibration can be performed at a later stage.

---

*For a possible improvement see Ref. [56, 57].

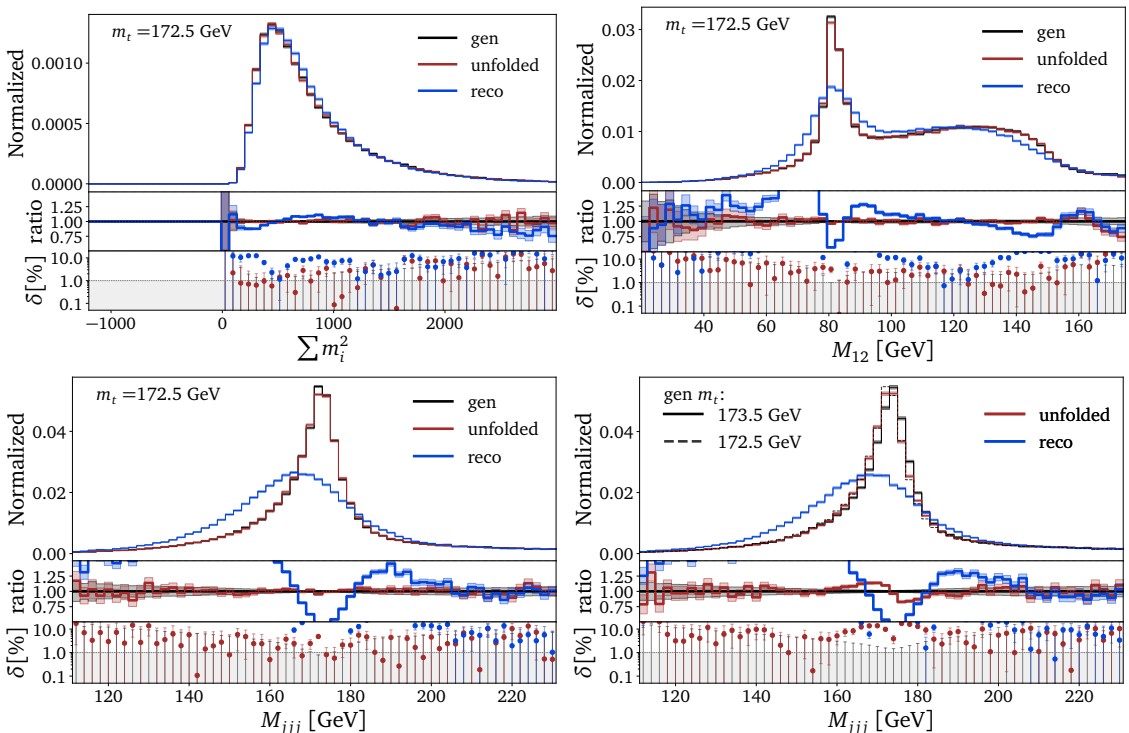

Figure 4: Kinematic distributions from the 4-dimensional unfolding. We also show the reco-level and the gen-level truth for $m_t = 172.5$ GeV. In the bottom-right panel we compare $M_{jjj}$ for $m_t = 172.5$ GeV to generated unfolding for $m_t = 173.5$ GeV, not seen during training.

Our unfolding setup follows Sec. 2.3. From Eq.(6) we know that we can extract the 3-jet mass as a proxy for the top mass from the set of single-jet and 2-jet masses. Because the single-jet masses are largely universal and not a good handle on the jet energy calibration, our first choice is to measure the top mass from a 4-dimensional unfolding of

$$\left\{ M_{j1j2}, M_{j2j3}, M_{j1j3}, \sum_i m_i \right\}. \tag{19}$$

The results are shown in Fig. 4. First, we see that we can unfold the sum of the single jet masses extremely well, with deviations of the unfolded data from the generator truth at the per-cent level. This means that we expect to be able to extract the 3-jet mass essentially from the sum of all 2-jet masses with a known and controlled offset.

Next, we show a 2-jet mass, with the characteristic $W$ peak and the shoulder at $m_{bj}^{\max}$. The $W$ peak is washed out at the reco-level, but the generative unfolding reproduces the gen-level extremely well. The relative deviation of the unfolded to the truth 2-jet mass distributions is at most a few per-cent, with no visible shift around the $W$ peak. The same quality of the unfolding can be observed in the $M_{jjj}$ distribution, perfectly reproducing the top mass at $m_t = 172.5$ GeV, the correct value in the training data and in the data which gets unfolded.

The problem with measuring the top mass from unfolded data appears when we unfold data simulated with a different top mass. In the lower-right panel of Fig. 4 we show the un-folded $M_{jjj}$ distribution for reco-level data generated with $m_t = 173.5$ GeV, unfolded with generative networks trained on $m_t = 172.5$ GeV. We see that the top peak in the unfolded data is dominated by the training bias of the network, specifically a maximum at $M_{jjj} = (172 \pm 1)$ GeV. This means the top peak is entirely determined by the training bias and hardly impacted by the reco-level data which we unfold.

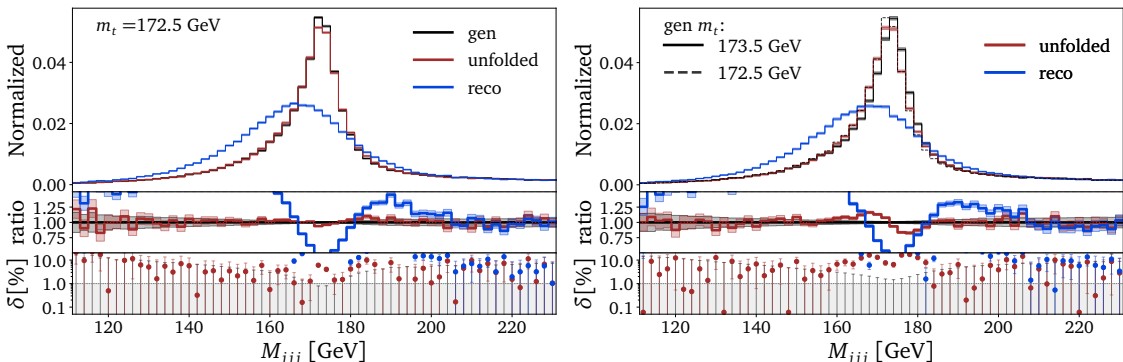

Figure 5: Kinematic distributions from 6-dimensional unfolding. In the right panel we compare $M_{jjj}$ for $m_t = 172.5\,\text{GeV}$ to generated unfolding for $m_t = 173.5\,\text{GeV}$, not seen during training.

From the 4-dimensional unfolding we know that the network learns the $W$ peak in the 2-jet masses and the top peak in the 3-jet mass at a precision much below the physical particle widths. The problem is that the bias from the network training completely determines the position of these mass peaks in the unfolded data. To confirm that these findings are not an artifact of our reduced phase space dimensionality, we repeat the same analysis for the 6-dimensional phase space

$$\left\{ M_{j_1 j_2}, M_{j_2 j_3}, M_{j_1 j_3}, m_{j_1}, m_{j_2}, m_{j_3} \right\}. \tag{20}$$

The unfolded 3-jet mass distributions are shown in Fig. 5, corresponding to the 4-dimensional case in Fig. 4. While the unfolded peak in $M_{jjj}$ is a bit worse than for the easier 4-dimensional case when unfolding the same value of $m_t$ as used in the training, the bias from the training remains in spite of the fact that we are weakening the expressive power of the unfolding network by adding distributions that are mildly affected by the peak position.

Finally, it is instructive to study the true and learned migrations between the reco-level and the gen-level 3-jet distribution. These are shown in Fig. 6, where in the left panel we see that the forward simulation maps the sharp peak at gen-level to a broader peak at reco-level. The problem with the central ellipse describing this physical migration by detector effects is that it

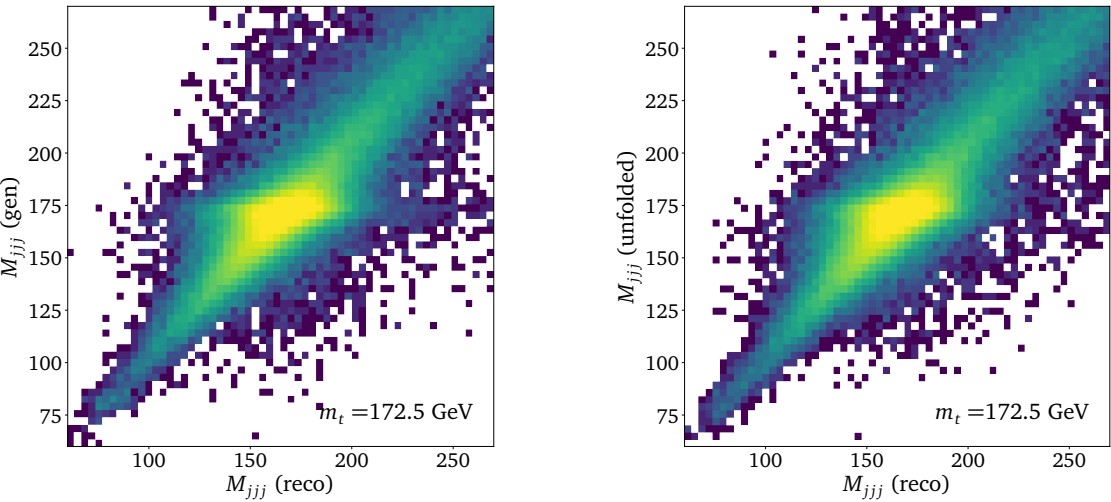

Figure 6: True and learned migrations in the $M_{jjj}$ distribution between reco-level and gen-level.

does not indicate any correlation between the $M_{jjj}$-values at reco-level and at gen-level. The learned migration in the right panel reproduces the forward migration exactly.

For the generative unfolding this means that small differences at reco-level will always be unfolded to the same sharp region at gen-level, independent of the information contained in the reco-level data. Following Sec. 2.4 and Eq.(16) the unfolded distribution $p_{\text{unfold}}(x_{\text{gen}})$ is entirely determined by the training choice $m_s$ and shows practically no dependence on the value $m_d$ encoded in the actual data.

## 3.2 Taming the training bias

The next question is how we can improve the situation where, $m_s$ being the top mass value used for the simulation and $m_d$ the actual top mass in the data, Eq.(16) turns into

$$
\begin{array}{ccc}
p_{\text{sim}}(x_{\text{gen}}|m_s) & & p_{\text{unfold}}(x_{\text{gen}}|m_s, \cancel{m_d}) \\
\Big\downarrow {\scriptstyle p(x_{\text{reco}}|x_{\text{gen}})} & & \Big\uparrow {\scriptstyle p_{\text{model}}(x_{\text{gen}}|x_{\text{reco}}, m_s)} \\
p_{\text{sim}}(x_{\text{reco}}|m_s) & \xrightarrow{\text{correspondence}} & p_{\text{data}}(x_{\text{reco}}|m_d).
\end{array}
\tag{21}
$$

In the unfolded distribution, the training information $m_s$ completely overwrites $m_d$. More-over, even if there was enough sensitivity, a classifier comparing two shifted mass peaks learns weights far away from unity, leading to numerical challenges. This means we cannot use the usual iterative methods to remove the bias from the training data.

Following the strategy from Sec. 2, we first increase the sensitivity on $m_d$. For this, we pre-process the data such that $m_d$ is directly accessible by adding an estimator of $m_d$ to the representation of $x_{\text{reco}}$. Ideally, this estimator would be inspired by an optimal observable. Such a one-dimensional observable with sufficient statistical precision should exist, and we know how to construct it. For the top mass we just use the weighted median of the 3-jet masses at reco-level, $M_{jjj}^{\text{batch}} = \frac{1}{N_{\text{batch}}} \sum_i^{N_{\text{batch}}} M_{jjj,i}$, where the sum runs over all events in one batch. For a batch size around $10^4$ events, this information will be strongly correlated with the top mass,

$$
M_{jjj}^{\text{batch}} \approx m_d \equiv m_t \Big|_{\text{data}}.
\tag{22}
$$

This batch-wise kinematic information can be extracted at the level of the loss evaluation, and it goes beyond the usual single-event information, similar to established MMD loss modifications of GAN training [15, 24].

Second, we weaken the bias from the training data by combining training data with dif-ferent top masses, but without an additional label,

$$
m_t = \{169.5, 172.5, 175.5\} \text{ GeV} \qquad \text{(combined training)}.
\tag{23}
$$

It turns out that it is sufficient to cover a range of top masses with separate, unmixed training batches. The range ensures that top masses in the actual data are within the range of the training data. We ensure a balanced training by enlarging the event samples with $m_t = 169.5$ and 175.5 GeV to match the size of the largest sample. This is done by repeating and shuffling the input data, which effectively uses these events several times per epoch. we avoid over-fitting using an appropriate regularization. The limited number of simulated events for the eventual analysis makes this training strategy sub-optimal. We expect larger and additional

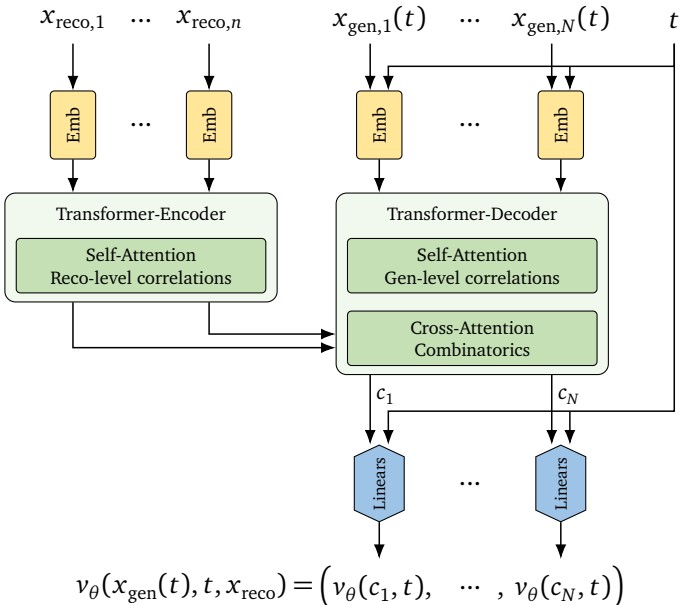

Figure 7: Schematic representation of a parallel transfusion network, adapted from [3].

$m_t$ simulations, unavailable at this time, to improve the results. We observed that both steps need to be included to ensure precise, unbiased results.

Obviously, this strategy of strengthening the dependence on $m_d$ and reducing the training bias is not applicable to all problems, and it does not lead to the endpoint of the Bayesian iterative method, but for our combined inference-unfolding strategy it works, and this is all we need.

**Transfusion architecture**

As the network task becomes significantly more difficult we replace the simple dense architecture with a transfusion network, described in detail in Refs. [3, 58] and visualized in Fig. 7.

Each component of the $n$-dimensional condition as well as of the time-dependent $N$-dimensional input $x(t)$ are individually embedded by concatenating positional information and zero padding. The embedded conditions are passed through the encoder part of a transformer, while the embedded input is passed through the decoder counterpart. In both transformer parts, we apply self-attention to learn the correlations in the condition and in the input. The network is complemented by a cross-attention between encoder and decoder outputs, to learn the correlations between conditions and inputs. These are crucial for the unfolding task. For every component of the input, the transformer returns one high-dimensional embedding vector $c_i$, which is mapped back to a one-dimensional component of the velocity field by a shared dense linear network. This way, we express the learned $N$-dimensional velocity field of Eq.(14) as

$$v_\theta(x_{\text{gen}}(t), t, x_{\text{reco}}) = (v_\theta(c_1, t), \dots, v_\theta(c_N, t)). \tag{24}$$

The hyperparameters of the network can be found in Appendix A.

Using the transfusion network we unfold the 4-dimensional phase space from Eq.(19). The results are shown in Fig. 8 (top row). We unfold data generated with two different top masses, $m_t = 171.5$ and $173.5$ GeV. Neither of these two values are present in the training data. We observe in both cases that the top mass as the main kinematic feature is reproduced

well, without a significant deviation from the gen-level distributions. The fitted peak values of the distributions are $m_{\text{peak}} = (172 \pm 1)\,\text{GeV}$ when unfolding data with $m_t = 171.5\,\text{GeV}$, and $m_{\text{peak}} = (174 \pm 1)\,\text{GeV}$ when unfolding data with $m_t = 173.5\,\text{GeV}$. While the bias might not have vanished entirely, it is well contained within the numerical uncertainties. We will extract the unfolded top mass value properly in Sec. 3.3.

**Dual network**

Given the more complicated training task, we observe a drop in performance when we increase the dimensionality to unfold the 6-dimensional phase space

$$x = \big( \{m_i\}, \{M_{ik}\} \big), \tag{25}$$

defined in Eq.(20) using the transfusion network. Inspired by Refs. [25, 26], we factorize the phase space density into two parts, each encoded in a generative network: the first network learns the individual jet mass directions in phase space, which are universal and do not depend on the value of $m_t$; the second network generates the 2-jet masses conditioned on the individual jet masses,

$$p(x_{\text{gen}}|x_{\text{reco}}) = \underbrace{p\big( \{m_{i,\text{gen}}\} \big| x_{\text{reco}}, M_{jjj}^{\text{batch}} \big)}_{\text{network 1}} \underbrace{p\big( \{M_{ik,\text{gen}}\} \big| \{m_{i,\text{gen}}\}, x_{\text{reco}}, M_{jjj}^{\text{batch}} \big)}_{\text{network 2}}. \tag{26}$$

Both CFM-transfusion networks also receive $M_{jjj}^{\text{batch}}$ calculated for a full batch using Eq.(6). For the event generation we first generate the unfolded jet masses $\{m_i\}$, pass them as a condition to the second network, and then generate the unfolded 2-jet masses $\{M_{ik}\}$.

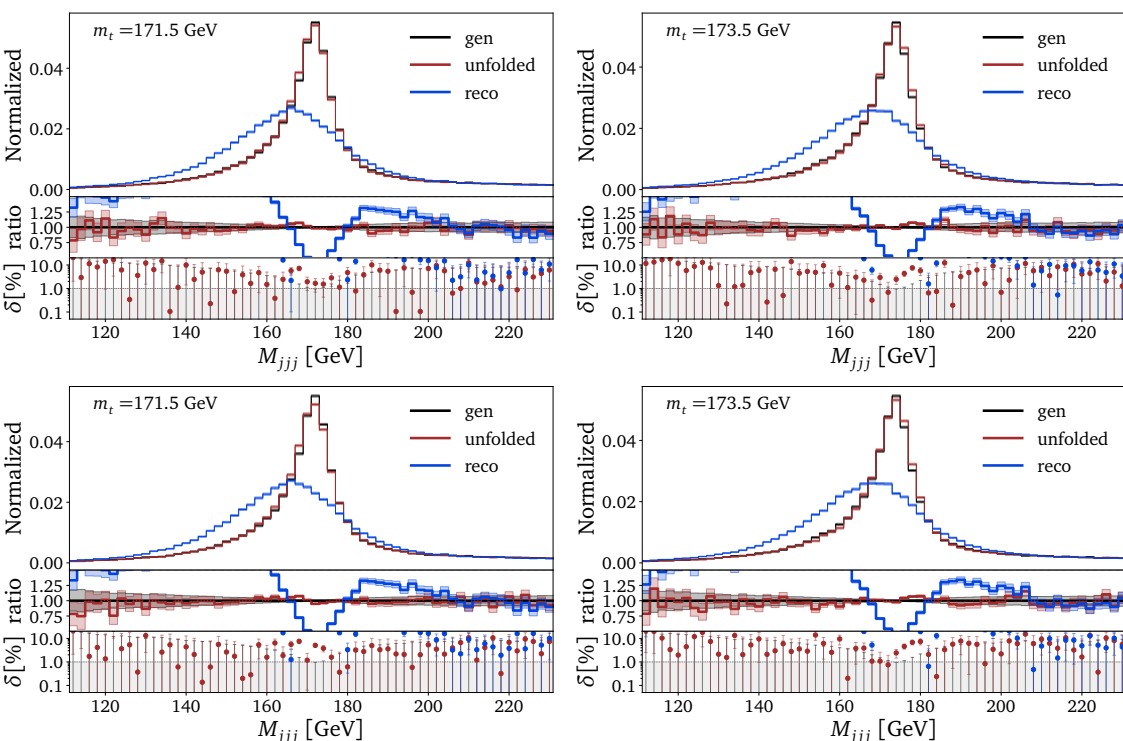

Figure 8: $M_{jjj}$-distributions from the 4-dimensional (top row) and 6-dimensional (bottom row) unfolding of data with $m_t = 171.5\,\text{GeV}$ (left column) and $m_t = 173.5\,\text{GeV}$ (right column). We train the network combining samples with three top masses, Eq.(23).

Looking at the 6-dimensional correlation giving $M_{jjj}$ in Fig. 8 (bottom row), we observe a hardly visible drop in performance, but still no bias from the training data. As before, we observe peak values at $m_{\mathrm{peak}} = (172 \pm 1)\,\mathrm{GeV}$ when unfolding data with $m_t = 171.5\,\mathrm{GeV}$ and at $m_{\mathrm{peak}} = (174 \pm 1)\,\mathrm{GeV}$ when unfolding data with $m_t = 173.5\,\mathrm{GeV}$.

## 3.3 Mock top quark mass measurement

We estimate the benefit from generative unfolding by repeating the top quark mass measurement from Ref. [34], but with a large number of bins in the $M_{jjj}$ histogram. The top mass is extracted from the binned unfolded distributions using a fit based on $\chi^2 = d^T V^{-1} d$, where $d$ is the vector of bin-wise differences between the normalized unfolded distribution and the normalized prediction from the simulated data. The covariance matrix $V$ contains the uncertainties and corresponding bin-to-bin correlations. A parabola fit provides the central value of $m_t$ and the standard deviation.

**Statistical and model uncertainties**

First, this fit requires the covariance matrix describing statistical uncertainties [59]. We sample $N$ times from the latent space, conditional on the reco-level events. This means we generate $N$ unfolded distributions from the posterior $p_{\mathrm{model}}(x_{\mathrm{gen}}|x_{\mathrm{reco}})$. We then use a Poisson bootstrap, where we assign a weight from a Poisson distribution with unit mean. The size of one replica is 52,000 events, corresponding to the approximate number of real data events. The number of events follows a Poisson distribution, with the mean given by the nominal sample size.

For the measurement, we create $N_{\mathrm{rep}} = 1000$ replicas by selecting the nominal number of reco-level events from the test dataset with $m_t = 172.5\,\mathrm{GeV}$ and the full datasets for the simulations at different top masses. We unfold each replica, calculate $M_{jjj}$, and use the histogram entries $u_i^{(n)}$ to compute the correlation matrix of statistical fluctuations as

$$
\mathrm{cov}_{ij} = \frac{1}{N_{\mathrm{rep}}} \sum_{n=1}^{N_{\mathrm{rep}}} (u_i^{(n)} - \bar{u}_i)(u_j^{(n)} - \bar{u}_j) \qquad \text{with} \qquad \bar{u}_i = \frac{1}{N_{\mathrm{rep}}} \sum_{n=1}^{N_{\mathrm{rep}}} u_i^{(n)}
$$

$$
\rho_{ij} = \frac{\mathrm{cov}_{ij}}{\sqrt{\mathrm{cov}_{ii}}\sqrt{\mathrm{cov}_{jj}}} \ . \tag{27}
$$

This procedure also takes into account the uncertainties due to the statistical fluctuations of $M_{jjj}^{\mathrm{batch}}$. The training of the network itself introduces correlations which are at least one order of magnitude smaller and therefore ignored in the measurement.

The $5 \times 5$ and $60 \times 60$ correlation matrices $\rho_{ij}$ from the 4-dimensional unfolding using the largest sample generated with $m_t = 172.5\,\mathrm{GeV}$ are shown in Fig. 9. We see two distinct sources of bin-to-bin correlations. In general, an event migrating from bin $i$ to bin $j$ gives rise to negative correlations in $\rho_{ij}$ between the two bins. Additionally, unbiasing the unfolding ensures that a shift in the batch-wise condition also shifts the unfolded peak. This effect, accounted for in the bootstrapping method, introduces an additional contribution to the bin-to-bin correlations. It causes positive correlations between bins on the same side of the peak and anti-correlations otherwise. In our case, both effects are most apparent in the peak region and its neighboring bins.

We follow Ref. [34] to estimate the uncertainty from the choice of $m_t$ in the simulation used for the unfolding. We evaluate the difference in each bin $i$ between the unfolded distribution and the corresponding simulated gen-level distribution. From the differences $d_i$, we construct

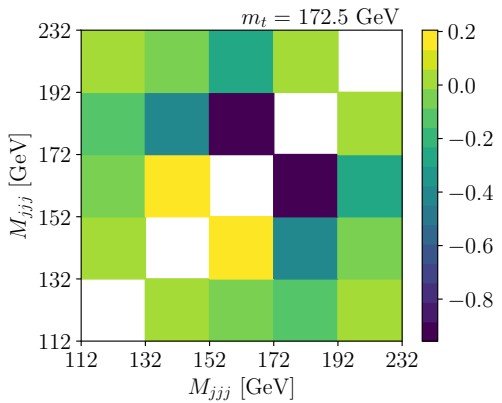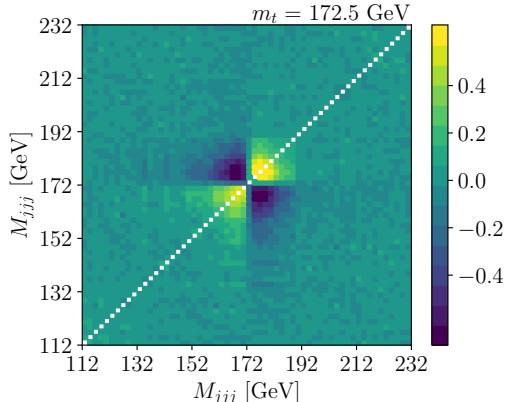

Figure 9: Correlation matrices obtained from $N_{\text{rep}} = 1000$ replicas for 5 bins (left) and 60 bins (right) in the 4-dimension unfolding with $m_t = 172.5\,\text{GeV}$.

a covariance matrix

$$\text{cov}_{ij}^{\text{model}} = \rho_{ij} d_i d_j , \tag{28}$$

where $\rho_{ij}$ are the correlations between bins $i$ and $j$. Because the bin-to-bin correlations are not known and we do not observe any systematic pattern, we choose a diagonal covariance matrix with $\rho_{ij} = 1$ for $i = j$ and $\rho_{ij} = 0$ otherwise. It was verified that other choices do not alter the results. To estimate the impact of this model uncertainty, we perform the $m_t$ extraction twice. First, we only include the statistical covariance matrix corresponding to 52,000 available events at the reco-level. Second, we repeat the same measurement also including the model uncertainty.

**Improvement**

To compare our new unfolding technique to the existing TUnfold results [34], we repeat the extraction using the simulated data set with 172.5 GeV and using the statistical covariance matrix from the measured data, published in HEPData [60]. The $\chi^2$-curves and the corresponding

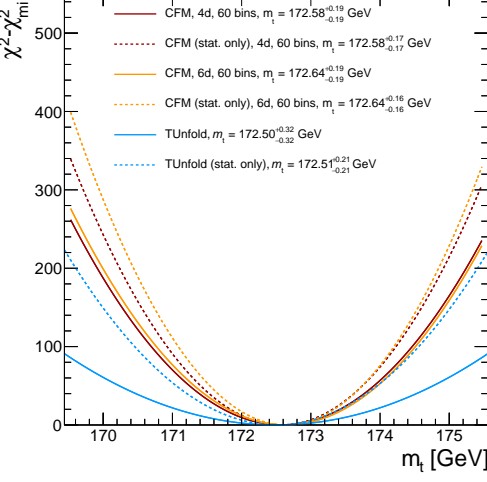

Figure 10: Extraction of $m_t$ with a $\chi^2$ test. The dotted lines include only statistical uncertainties, while the solid lines also include the model uncertainty from the choice of $m_t$.

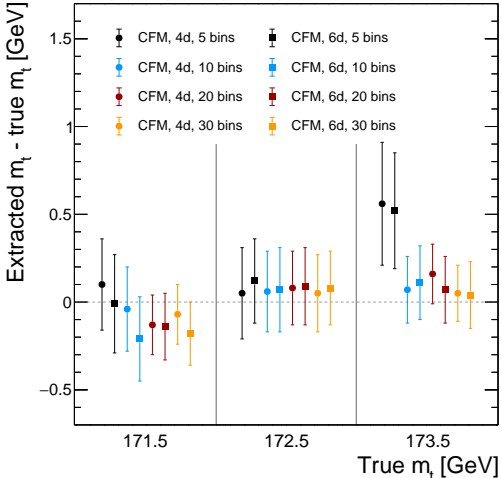 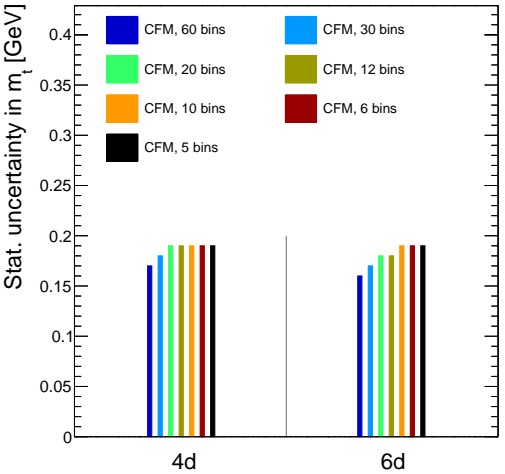

Figure 11: Deviation of the extracted top mass from the reco-level truth, employing 4-dimensional and 6-dimensional unfoldings with different numbers of measurement bins for $m_t = 171.5$, $172.5$, and $173.5$ GeV (left) . The size of the statistical uncertainties in $m_t$ from the 4-dimensional and 6-dimensional unfoldings with different binnings, assuming $m_t = 172.5$ GeV (right).

results are displayed in Fig. 10, where we show the 4-dimensional and 6-dimensional unfoldings with 60 bins and the TUnfold result. We see that the uncertainty in the choice of $m_t$ is reduced from being a leading model uncertainty in the CMS measurement to a much smaller level. In addition, the statistical uncertainty is reduced because of the finer binning of the unbinned unfolding.

To confirm that the choice in $m_t$ does not leave a residual bias, we repeat the top quark mass extraction for unfolded data obtained from reco-level data simulated with different top masses. The results are shown in the left panel of Fig. 11. For a top mass of $m_t = 173.5$ GeV, we observe a bias of about 0.5 GeV when using a measurement with 5 bins. This is not surprising as the exact binning has been optimized for a minimal model dependence in the CMS measurement, which we did not do here. While the bin width in the unfolding with TUnfold is limited by the jet mass resolution, we test various binning schemes for the unbinned unfolding. The bias gets reduced when using more bins in the measurement, as expected because the binning introduces a regularization in the unfolding which leads to a model dependence. With 10 and more measurement bins, we observe that the bias from the model dependence is removed. For more measurement bins than 60, the comparably coarse grid of gen-level distributions with $m_t = \{169.5, 171.5, 172.5, 173.5, 175.5\}$ GeV leads to an unstable closure test.

To circumvent this limitation, we interpolate the gen-level distributions for $m_t$-values close to 172.5 GeV, where three samples with a separation of 1 GeV are available and a linear dependence of the bin content as a function of $m_t$ represents a valid approximation. Now, we can compare the resuling values of $m_t$ from the generative unfolding with 5 to 60 bins in terms of the statistical uncertainty. The results are displayed in the right panel of Fig. 11, indicating an increase in the statistical precision in $m_t$ due to the improved resolution.

## 3.4 Full phase space unfolding

As a last step of our unfolding program, we unfold the full 12-dimensional phase space given the measured top mass. This has the advantage that the leading source of training bias is removed. Following the same precision arguments as before, we keep the mass basis of Eq.(20)

for the first 6 of the 12 phase space dimensions. This ensures that the 2-jet and 3-jet masses are reproduced well, albeit not at the level of the dedicated first unfolding step.

The remaining phase space dimensions are

$$x = \left( \{m_i\}, \{M_{ik}\}, \{p_{T,i}\}, \{\eta_i\} \right) \qquad i, k = 1, 2, 3 , \tag{29}$$

all other kinematic observables can be computed from these basis directions. For the 12-dimensional unfolding we use a single transfusion network, after checking that the dual network does not lead to an improvement. The hyperparameters are given in Appendix A. Two kinematic distributions are shown in Fig 12. In the left panel, we see that the top mass peak is learned almost as well as for the 4-dimensional and 6-dimensional cases. Indeed, this is the case for all jet masses and 2-jets masses, which are combined to the 3-jet mass with the top peak.

A serious issue arises from the azimuthal angle between the two leading jets, $|\Delta\phi_{12}|$. According to Eq.(4) this angle is learned as a correlation of 7 phase space directions. Moreover, we do not have access to the azimuthal angles, only to the cosine of differences between angles. Here the problem arises that the network does not ensure that this cosine comes out in the physical range $-1 \dots 1$. We enforce the physical range by clipping the cosine for small angles to one, which causes a mis-modelling of the small-$|\Delta\phi_{12}|$ regime, shown in the right panel of Fig. 13.

A simple way to improve this mis-modelling is to require $\cos\Delta\phi_{12} < 1$. However, from Fig. 12 we know that this does not solve the problem. Instead, we accept the fact that for unfolding the masses well we might have to pay a prize in the coverage of the angular correlations, and we apply an additional acceptance cut

$$\Delta\phi_{ik} > 0.1 \tag{30}$$

both, at the reco- and gen-levels in our simulated events. This reduces the size of the unfolded dataset by 30%. An extended set of unfolded kinematic distribution after this cut are shown in Fig 13.

We know that our unfolding method covers correlations between the original phase space directions well, because many of the kinematic observables shown in Fig 13 are built from complex correlations of our phase space basis. However, to end with a nice figure and to drive home the message that high-dimensional unfolding using conditional generative networks does learn the corresponding correlations well, we show one of our favorite correlations in Fig 14. Indeed, there is literally no difference in the correlations between two of the three

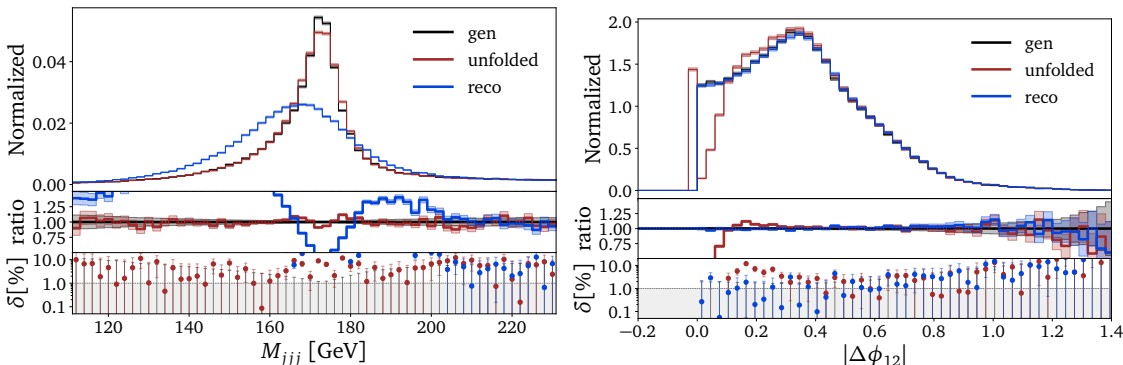

Figure 12: Kinematic distributions from full, 12-dimensional unfolding. We show the 3-jet mass as well as the azimuthal angle between the two leading jets.

2-jet masses. This correlation also confirms that the condition $M_{ik} \approx m_W$ leads to three distinct lines in phase space, where close to the crossing points it is impossible to reconstruct which two of the jets come from the $W$ decay.

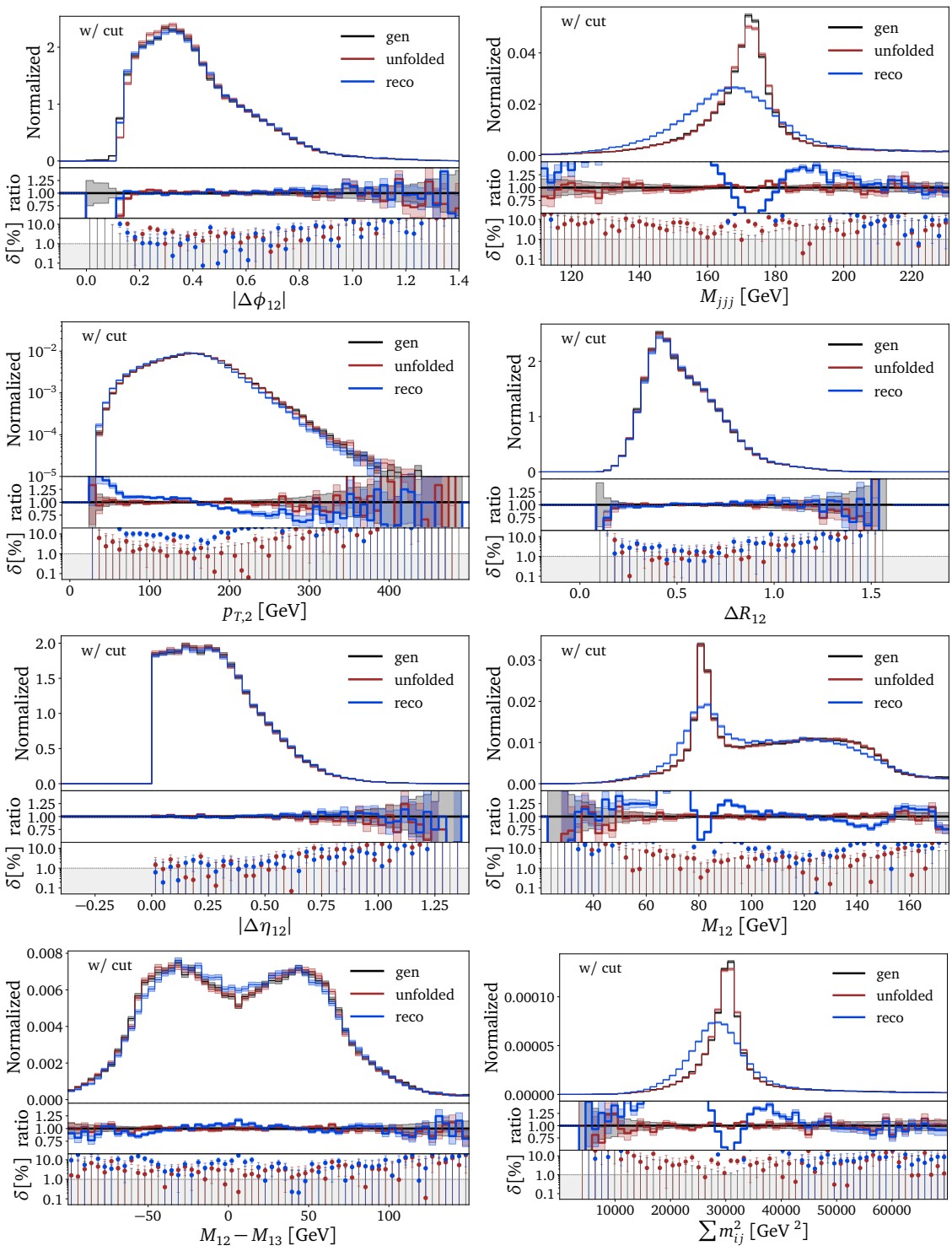

Figure 13: Kinematic distributions from full, 12-dimensional unfolding. We show the target 3-jet distribution, the azimuthal angle between the jets after cut, and a set of single-jet observables, 2-jet correlations, and 3-jet correlations (top to bottom).

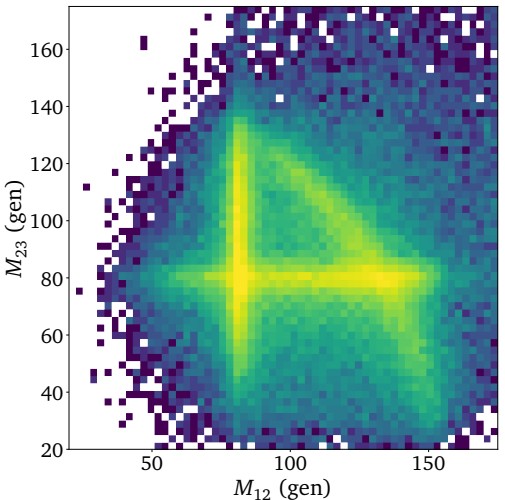 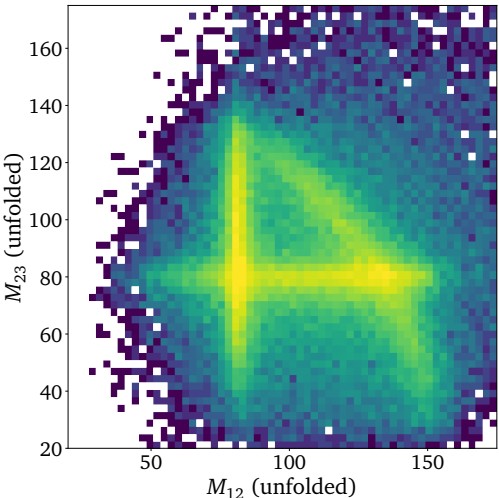

Figure 14: Correlation of two 2-jet masses at gen-level truth (left) and after unfolding (right).

## 4   Outlook

Unfolding is one of the ways modern machine learning is transforming the way we can do LHC physics. Employing an inverse simulation, it allows for the efficient analysis of LHC data by the LHC collaborations, to combine analyses between different experiments, and even make unbinned, unfolded data accessible to researchers outside the experimental collaborations. Unfolding has been used in particle physics frequently, but modern neural networks allow us to unfold a high-dimensional phase space without a choice of binning. This technical advance will turn multi-dimensional and unbinned unfolding into a standard analysis method at the LHC and future experiments.

For our study, we unfold detector effects from boosted top quark decay data using state-of-the-art conditional generative networks. Unfolding decay kinematics is especially challenging because we expect a large model dependence and even systematic bias from the choice of the top mass in the simulated training data. Our study shows that generative unfolding can solve this problem and provides a first milestone towards incorporating generative unfolding in an existing LHC analysis.

First, we showed that for an appropriate phase space parametrization, a combination of diffusion network and transformer can reliably unfold a 4-dimensional and 6-dimensional subspace of the full top-decay phase space at the percent level precision. This included the 3-jet mass as a proxy to the top mass. The problem in this unfolding is a strong bias from the top mass used to generate the training data. To compensate this bias we added a global estimate of the top mass to the representation of the measured data and weakened the training bias by including a range of top masses there. As a result of these two structural modifications, the top mass bias was essentially removed.

Using this setup we showed how to extract the top mass along the lines of a recent CMS analysis [34]. We included two covariance matrices, one describing all statistical uncertainties and one covering the model uncertainty from the training data. We found that, indeed, the impact of the model uncertainty is becoming irrelevant, and that the error in the top mass can be reduced when using the kind of fine binning allowed by the unbinned unfolding method.

Finally, we unfolded the full, 12-dimensional phase space for a given top mass. One failure mode in reproducing the angular distributions was induced by our phase space parametrization. However, a simple lower cutoff on the azimuthal angular separations of the top decay

jets allowed for an excellent reproduction of all correlations.

This study is meant to serve as a blueprint for an actual CMS analysis, both, for a top mass measurement and for a wider use of the unfolded data.

## Acknowledgements

Most importantly, we would like to thank the organizers and experts at the 2024 Terascale Statistics School for pointing out that nobody in their right mind would ever attempt to use unfolding for a mass measurement. We completely agree with that highly motivating point of view.

Moreover, we like to thank Henning Bahl, Anja Butter, Theo Heimel, Nathan Huetsch and Nikita Schmal for many valuable discussions, and Andrea Giammanco and Anna Benecke for useful discussions on the Delphes detector simulation. This research is supported through the KISS consortium (05D2022) funded by the German Federal Ministry of Education and Research BMBF in the ErUM-Data action plan, by the Deutsche Forschungsgemeinschaft (DFG, German Research Foundation) under grant 396021762 – TRR 257: *Particle Physics Phenomenology after the Higgs Discovery*, and through Germany's Excellence Strategy EXC 2181/1 – 390900948 (the *Heidelberg STRUCTURES Excellence Cluster*). We would also like to thank the Baden-Württemberg Stiftung for financing through the program *Internationale Spitzenforschung*, project *Uncertainties – Teaching AI its Limits* (BWST_ISF2020-010). LF is supported by the Fonds de la Recherche Scientifique - FNRS under Grant No. 4.4503.16. SPS is supported by the BMBF Junior Group Generative Precision Networks for Particle Physics (DLR 01IS22079). The research work of DS has been funded by the Austrian Science Fund (FWF, grant P33771). The authors acknowledge support by the state of Baden-Württemberg through bwHPC and the German Research Foundation (DFG) through grant no INST 39/963-1 FUGG (bwForCluster NEMO).

## A  Hyperparameters

| Parameter | |
|---|---|
| LR sched. | cosine |
| Max LR | $10^{-3}$ |
| Optimizer | Adam |
| Batch size | 16384 |
| Network | Resnet |
| Dim embedding | 64 |
| Intermediate dim | 512 |
| Num layers | 8 |

Table 1: Parameters for the 4-dimensional and 6-dimensional networks in Sec. 3.1.

| Parameter | 4D | 6D |
|---|---|---|
| Epochs | 800 | 500(+1000) |
| LR sched. | cosine | cosine |
| Max LR | $10^{-3}$ | $10^{-3}$ |
| Optimizer | Adam | Adam |
| Batch size | 10000 | 10000 |
| Dropout | 0.1 | 0.1 |
| Network | Transfusion | Transfusion |
| Dim embedding | 64 | 64 |
| Intermediate dim | 512 | 512 |
| Num layers | 4 | 4 |
| Num heads | 4 | 4 |

Table 2: Parameters for the 4-dimensional and 6-dimensional networks in Sec. 3.2.

| Parameter | 12D |
|---|---|
| Epochs | 500 |
| LR sched. | cosine |
| Max LR | $10^{-3}$ |
| Optimizer | Adam |
| Batch size | 16384 |
| Dropout | 0.1 |
| Network | Transfusion |
| Dim embedding | 128 |
| Intermediate dim | 512 |
| Num layers | 6 |
| Num heads | 4 |

Table 3: Parameters for the 12-dimensional network in Sec. 3.4.

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
