# Peer review of "How to Unfold Top Decays"

_SciPost Physics_

## Round 2 · Referee Report · Anonymous (Referee 1) · 2025-4-1

Strengths

This paper studies the application of unbinned unfolding methods to semileptonic top-antitop final states in a boosted topology. These unbinned methods use ML algorithms to learn the features of the multidimensional phase space and allow the detector effects to be removed from the data.

(1) the authors demonstrate (for the first time) that the hadronic top-decay can be unfolded at the per-event level, allowing the mass of the top decay products to be reconstructed and the top quark mass to be extracted by a fit to a newly-binned distribution.

(2) The authors carry out detailed and robust studies into the unfolding performance, including the underlying ML architecture, methods to reduce biases from the assumed top quark mass in the simulation/training.

(3) The authors compare the performance of their unfolding when extracting the top mass to the results published CMS data

(4) The results demonstrate that the method works as a proof of principle and this opens up future possible applications to LHC data.

Weaknesses

(1) The paper does not address the impact of backgrounds, which in particular could bias a constraint on the trijet mass sampled from the 'data' in the training. This is discussed below in the requested changes

(2) Likewise the paper does not address the impact of additional sources of systematic uncertainty . Of particular interest is the possible interplay between these systematics and the top mass used in the training. This is discussed below in the requested changes

Report

I recommend the paper is published after the authors answer the questions detailed below and modify the draft accordingly. The research done in this paper meets the SciPost criteria for originality.

Requested changes

(1) Backgrounds: W+jets and single-top are present in the CMS phase space with about 5% contamination from each (see ref [34]). The paper does not address backgrounds and there are some questions that they pose: - Most importantly for the results in this paper: could backgrounds bias the m_jjj^batch requirement in sec 3.2? This is particularly relevant for the W+jets background, which will have a different shape, but might also be true for single top. The paper should explain how this bias would be mitigated during training. Ideally, this bias could be studied in this paper along with any solutions needed to mitigate it. - How can backgrounds be subtracted in this method? I presume this has been studied elsewhere and if so should be cited. If not, some statement needs to be made as to how it can be done.

(2) Systematics: the paper considers the bias due to the choice of top mass in the training and takes steps to mitigate the bias. However, there are systematic uncertainties that change the shape of the m_jjj distribution. During unfolding, there would presumably be some interplay between these systematics and the steps taken to remove the top mass bias. It isn’t clear whether any impact of these systematics is then smaller, similar to, or larger than the systematics in standard unfolding methods using TUnfold. Systematics that jump to mind include jet energy scale/resolution and the hadronisation/shower models. At minimum, some statements are needed to explain that these systematics exist and qualitatively explain the impact (perhaps by citation to previous work). Ideally, the authors could perform some sort of injection test to show the impact of such systematics on this method.

(3) Comparison to CMS data: Fig 10 shows that the statistical precision is much better for the unbinned unfolded method compared to TUnfold and this is stated to be from the finer binning allowed in the analysis. I think the discussion around this needs to be more detailed: - The improvement looks better than simply the finer binning, because the TUnfold stat-only error in fig 10 is +-0.21 whereas the 5-bin CFM fit has a stat-only error of +-0.19 and the CFM-4d 60 bin result has a stat-only error of +-0.17. This feature should be explained in the text. - It is likely that this difference is due to the use of the CMS measurement, which contains background subtraction and also fluctuations in the data itself. Would it not be better to compare apples-to-apples by applying TUnfold directly to your simulated events? - If there is an improvement in stat uncertainty die to finer binning, there might be some tradeoff with worse systematics due to jet energy resolution. This should be discussed. - More trivially, the y-axis range on figure 10 should be reduced as we are most interested in the 0 < Delta Chi^2 < 10.

(4) Results with and without mjjj sampling: on page 13, it is stated that both the mjjj batch sampling of data and the use of different top masses in the training are required for unbiased results. Could a plot be added to show this? ie showing original bias, inclusion of only mjjj batch sampling, inclusion of only combined training samples, inclusion of both. It would help the reader to understand the relative importance of each step.

Minor queries/changes:

(5) Does the unfolding rely on events being present at both truth and reco, or also correct for truth&!reco and reco&!truth?

(6) Text improvements: - finite efficiency -> inefficiency (p4) - recoconstruction -> reconstruction (p5) - section 2.4: this is aiming for a complete description of generstove unfolding, but quite a lot of terms are not defined, ie w(x_gen), w(x_reco), p_latent.

Recommendation

Ask for minor revision

---

## Round 2 · Referee Report · Anonymous (Referee 2) · 2025-4-15

Strengths

The paper presents a robust approach to tackling a well-known challenge in unfolding top-quark decay data.

1) The key notable strength in this paper is the detailed strategy presented for mitigating training bias. The authors take a thoughtful approach by combining samples produced with different assumed top masses and incorporating a batch-wise estimator (as described around Equation (22) and (23)) to lessen the influence of the simulation’s fixed input. This feature is particularly important in ensuring that the unfolded data better reflect the true underlying physics, rather than simply reproducing the bias of the training sample.

2) a second strength is not only the demonstration of how complex kinematic correlations can be preserved and measured with higher precision than conventional binned techniques but that this can be applied in a practical setting. This is a meaningful advance over the standard practices, which are typically limited to low-dimensional histograms and struggle with capturing intricate inter-variable correlations.

3) Finally the paper provides an in-depth discussion of the network architecture used (the conditional Flow Matching network combined with a transfusion network), complete with discussions of hyperparameters in the Appendix. This detailed exposition not only underpins the technical validity of the approach but also offers practical insight for experimental groups considering adopting similar techniques.

Weaknesses

I think much of my difficulty with reviewing this paper is that it attempts to tackle two goals at once. On the one hand, it develops a novel ML-based unfolding strategy with a specific unbiasing procedure and sophisticated network architecture. On the other, it aims to address phenomenological details specific to top-quark decays. However, on the methodological front, the paper mainly shows qualitative results—like ratio plots and comparisons of unfolded peak positions—without rigorous statistical bias analysis or extensive benchmarking against alternative ML methods such as OmniFold or conditional INNs. On the phenomenological side, while it does present results on unfolding a 12-dimensional phase space relevant to top decays, it stops short of a thorough treatment of experimental complexities like background contamination and systematic uncertainties. In this sense, the paper does not fully satisfy either ambition, as it leaves important quantitative and comparative questions open on both fronts.

1) My major gripe with this document is that paper’s demonstration of the benefits of unbinned, high-dimensional unfolding is based mostly on qualitative ratio plots and the comparison of peak positions. While the authors show that their approach can shift the unfolded peak closer to the true top mass when using different training samples, they do not provide a rigorous statistical analysis of bias or a detailed coverage study. This leaves some uncertainty about how well the method performs in practice compared to the underlying theoretical expectations.

2) The claim from the abstract that "generative unfolding enables both tasks and how both benefit from unbinned,high-dimensional unfolding" is highly suspect as 'both' in this context is just that unfolded data can be used for both measurements and anomaly detection, which is hardly under question and the benefits of high-dimensional unfolding have already been shown in numerous papers including those by some of the authors . The paper does not include a direct, comprehensive comparison against other modern ML‐based unfolding techniques. Although the generative method is compared to traditional binned methods like TUnfold in parts, the work falls short of juxtaposing its performance against approaches such as OmniFold or conditional invertible neural networks. This makes it harder to assess whether the proposed unbiasing step and the overall network design offer a genuine breakthrough over existing ML methods, simply an incremental refinement, or a choice in which users are asked to decide between the de-biassing effect introduced here or some feature of the other methods.

3) One final weakness is the dependency on the specific training strategy and simulation conditions—for instance, the approach of mixing samples with different assumed top masses and incorporating a batch-wise median estimator is sensible, but the paper does not fully explore the robustness of these choices under varying systematic uncertainties. I would expect such a study as part of an experimental measurement of the top-quark mass but it seems highly out of place here and doesn't seem to match with the core 'goal' of the paper. The results are largely presented in terms of ratios and bin differences, and the limited treatment of uncertainties—both statistical and systematic—raises the question of how the method would perform when applied to more complex or real-world data where the simulation may deviate even further from reality.

Report

I think this is a good paper and a good result and body of work. It should be published and despite all my complaints and recommendations I really don't think that there would be much work necessary to bring this to a suitable level.

...However...

My key point is the that authors must clarify, "what is the research question being answered here?" which is a question made particularly difficult by the excellent bodies of work already presented by some of the authors here on this very topic (usually in this very journal) in which many of the possible questions addressed by this document have already been addressed. In my mind more effort needs to be made to highlight what makes this result stand out.

  • if the goal is to showcase a novel methodology, then the results need to be more statistically rigorous, with less focus on this specific use case.

  • if the goal is to showcase the improvements to this specific use case, then this use case needs much much description justification, the results need to compare with other ML solutions, with additional focus on reconstruction techniques/backgrounds/systematics etc.

Requested changes

1) A specific comparison needs to be made against other ML unfolding algorithms - the one paragraph 'explanation' on page 8, whilst true, is still just speculative in this form regardless of "its mathematical foundation". You do not show that others fail or explain why and how they would, nor is this the key message. Simply run the analysis also using at least one other method and stick the results in the appendix or along side the TUnfold methods.

2) If you are making comparisons to TUnfold with respect to biases, the RooUnfold authors documented the accepted statistical procedure in https://arxiv.org/abs/1910.14654 this should be fairly trivial for you to implement and would strengthen the claims of this paper hugely.

Recommendation

Ask for minor revision

---

## Editorial Decision

resubmitted